# TAVI after More Than 20 Years

**DOI:** 10.3390/jcm12175645

**Published:** 2023-08-30

**Authors:** Adriana Postolache, Simona Sperlongano, Patrizio Lancellotti

**Affiliations:** 1Cardiology Department, GIGA Cardiovascular Sciences, University of Liège Hospital, CHU Sart Tilman, 4000 Liège, Belgium; adriana.postolache@gmail.com; 2Devision of Cardiology, Department of Translational Medical Sciences, University of Campania Luigi VanVitelli, 80131 Naples, Italy; sperlongano.simona@gmail.com

**Keywords:** TAVI, aortic stenosis

## Abstract

It has been more than 20 years since the first in man transcatheter aortic valve intervention (TAVI), and during this period we have witnessed an impressive evolution of this technique, with an extension of its use from non-operable patients to high-, intermediate- and even low-risk patients with aortic stenosis, and with a decrease in the incidence of complications. In this review, we discuss the evaluation of patients before TAVI, the procedure and the changes it has seen over time, and we present the current main complications and challenges of TAVI.

## 1. Introduction 

Since the first in man transcatheter aortic valve intervention (TAVI) performed by Dr. Alain Cribier in 2002 in a non-operable aortic stenosis (AS) patient, TAVI has changed the lives of so many patients for whom medical treatment was, up to then, the only option. During this 20 year period, the increased knowledge on pre-procedural planning, the important technological improvements in transcatheter valves, the increased experience and the numerous studies that have been carried out have permitted an expansion of the indications for TAVI, from inoperable patients to high- and intermediate-risk patients [1,2]. However, although more recent studies, have shown that TAVI is not inferior to surgical aortic valve replacement (SAVR) in low-risk patients, there are questions regarding long-term valve durability, in particular when it comes to using it in younger patients, the risk of embolic events and the need for pacemaker implantation. Furthermore, its use in patients with bicuspid aortic valves and questions regarding coronary artery access, in particular after valve-in-valve TAVI, represent current challenges to its wider use.

## 2. Patient Selection and Pre-Procedural Evaluation

The success of the intervention is dependent on patient selection. First, a detailed, stepwise, multiparametric and often multimodality evaluation of patients with AS is necessary for confirming the severity of AS and for evaluating the necessity of intervention. Echocardiography (transthoracic, stress and transesophageal echocardiography, in particular, with 3D) holds a central role in the diagnosis of severe AS, and it offers valuable information about systolic and diastolic left ventricular function, the presence of other valve diseases, of pulmonary hypertension or of right ventricular dysfunction [1,2,3]. These data offer important prognostic information, which go beyond AS severity, and should be taken into consideration in the heart team discussion [4,5]. In patients with discordant grading or when the severity is discordant with patient’s symptoms, the use of other imaging modalities such as cardiac computer tomography (CCT), cardiac magnetic resonance imaging, the use of cardio-pulmonary exercise testing or of biomarkers should be considered [1,2,3].

While echocardiography holds a central role in evaluating the severity of AS, CCT holds the key role for evaluating the feasibility of TAVI. The aortic annulus and aortic root dimensions, as well as the distance from the aortic annulus to the ostium of the coronary arteries, can be accurately measured during the cardiac cycle using electrocardiography-gated CT angiography [6,7]. This information enables the selection of the most suitably sized prosthesis, and thus contributes to a decrease in the risk of complications, such as significant paravalvular regurgitation or leak (PVL), annular rupture and coronary ostium obstruction. The presence and location of calcifications on the aortic valve, the aortic annulus and the aortic root also offer important prognostic information about the risk of PVL. Furthermore, the determination of the best fluoroscopic incidence for delivering the prosthesis can be extracted from the CT data set, and thus, CT can help in decreasing the use of contrast and radiation exposure during the procedure [6,7]. The CT mapping of the arterial vascular system, in particular, the evaluation of the presence of aorto-iliac calcifications, tortuosity and the measurements of vascular diameters, are particularly important for determining the most suitable vascular access site for delivering the prosthesis [6,7]. For these reasons, CCT angiography has become the standard imaging modality for evaluating the feasibility of TAVI. However, in patients with severe chronic kidney disease, 3D transesophageal echocardiography can be used instead, as studies have shown a good correlation between CCT and transesophageal 3D dimensions of the aortic annulus [8,9]. Angiography or vascular Doppler echocardiography can be used for vascular access site evaluation. 

After the pre-procedural evaluation, the patient is discussed in the heart team meeting, and the decision between SAVR and TAVI should be made on a case-by-case basis, taking into consideration the risk of the surgical intervention, the patient’s age and estimated life expectancy, the comorbidities and the presence of certain anatomical and procedural characteristics that could favor TAVI (feasible transfemoral TAVI, porcelain aorta, previous chest radiation, severe chest deformation, the presence of a coronary graft passing behind the sternum, or a high likelihood of severe patient–prosthetic mismatch) or SAVR (aortic annulus dimensions unsuitable for TAVI, high risk of coronary artery obstruction due to coronary ostia implantation < 10 mm from the annulus or heavy leaflet/left ventricular outflow calcifications, or the presence of bicuspid aortic valve, etc.) [1,2]. 

## 3. The TAVI Procedure in 2023 

There are two types of transcatheter valves, balloon-expandable and self-expandable.

The third generation of balloon-expandable SAPIEN^TM^ valves (Edwards Lifesciences Corporation, Irvine, CA, USA) includes the SAPIEN 3 and the SAPIEN 3 Ultra valves. They are composed of a cobalt–chromium cylindrical stent into which three symmetric leaflets made of bovine pericardium are mounted. They have a sealing skirt meant to decrease the risk of PVL. The short frame height and the open cell geometry of the SAPIEN 3 Ultra valve are meant to facilitate coronary access after TAVI. 

The most widely used self-expanding valve is the CoreValve^TM^ (Medtronic, Inc., Minneapolis, MN, USA), which consists of an asymmetrical, self-expanding nitinol frame, into which are mounted three leaflets of porcine pericardium. The more recent Evolut R, Evolut Pro and Evolut Pro + valves can be recaptured and repositioned after deployment, and the Evolut Pro and Evolut Pro + valves also have an outer pericardial wrap for decreasing PVL. The experience with other prostheses, such as the ACURATE TA (Symetis SA, Ecublens, Switzerland), the Direct Flow valve (Direct Flow Medical Inc, Santa Rosa, CA, USA), the Engager (Medtronic, Inc., Minneapolis, MN, USA), designed for apical access, and the JenaValve (JenaValve Technology GmbH, Munich, Germany), specifically designed for the treatment of aortic regurgitation, is more limited. There are little data regarding a direct comparison between the different prostheses. In a direct head-to-head comparison of the SAPIEN 3 balloon-expandable and the Evolut R self-expandable valves, the mortality rate was similar, but SAPIEN 3 had numerically lower rates of at least moderate PVL and primary pacemaker implantation, with a higher rate of stroke [10].

The procedure can be performed in the cardiac catheterization or in the hybrid operating room. Although, for many years, TAVI was performed under general anesthesia, with angiographic and transesophageal echocardiographic guidance, it is currently performed in most centers under conscious sedation and local anesthesia, with angiographic guidance only. In the ipsilateral leg, femoral arterial access is obtained for aortic angiography and a venous sheath is inserted, through which a temporary pacemaker is placed in the right ventricle. The contralateral artery is cannulated. Once the patient is anticoagulated, the aortic valve is crossed and a guidewire remains in place in the left ventricle. Then, the delivery sheath is inserted in the descending aorta. A balloon aortic valvuloplasty is performed under rapid ventricular pacing. The prosthesis is then advanced retrogradely to the level of the ascending aorta, and after confirmation of the appropriate location with angiography, the valve is deployed during rapid ventricular pacing. The transvalvular gradients are measured and the presence of PVL is evaluated. If significant PVL is present, post-dilatation is performed. The sheath is withdrawn with careful blood pressure monitoring and contrast administration at the iliac arteries, for identifying an eventual vascular complication, which should be treated promptly. A transthoracic echocardiography is performed at the end of the procedure to evaluate the function of the prosthesis, in particular the severity of aortic regurgitation, and to evaluate the presence of new wall motion abnormalities that could be related to coronary artery obstruction or the presence of pericardial effusion. A transthoracic echocardiography is performed before the patient leaves the hospital, and this serves as a comparative exam for the follow-up. 

In more recent years, a minimalist PCI-like TAVI procedure has been introduced. The main aspects of minimalist TAVI are the performance of the procedure under conscious sedation, sometimes without an anesthesiologist in the room, the use of percutaneous transfemoral access, the use of radial instead of femoral contra-lateral access, the use of left-ventricle guide-wire pacing instead of transvenous right-ventricular pacing, restricting the pre-dilatation of the valve only to selected cases, no intensive care unit monitoring after the procedure, and even same-day or next-day discharge [11,12,13]. In centers with good expertise, minimalist TAVI can be safely used in rigorously selected cases of transfemoral TAVI and can be associated with a decrease in the total hospital stay and of costs related to the hospitalization [11,12,13].

## 4. TAVI Complications and Current Challenges

Overall, the incidence of complications after TAVI has decreased significantly due to the increase in experience, the use of CCT as the main imaging modality for evaluating the feasibility of TAVI, the significant technological advancements in the design of the prostheses, and the decrease in the size of the sheaths. A summary comparison of the pivotal studies of TAVI in patients at different surgical risks is presented in Table 1, whereas Table 2 presents the incidence of the main TAVI complications in more recent trials.

### 4.1. Paravalvular Regurgitation or Leak

The incidence of PVL after TAVI has decreased significantly in the last two decades, due to the detailed pre-procedural evaluation with improvements in patient and prosthesis selection (avoiding under-sizing, recognizing the importance of severe valvular calcifications in predicting the risk of PVL), the technological advancements seen in the design of prosthetic valves and the increased experience. However, with the exception of the PARTNER 3 study, which showed similar rates of moderate-to-severe PVL in TAVI and SAVR, all other studies showed a higher incidence of PVL after TAVI as compared to SAVR, with 22–29% of patients having mild PVL and an incidence of moderate-to-severe PVL between 0.6–3.7% after balloon-expandable, and between 3.5–5.3% after self-expandable valves [14,15,16,17] (see Table 2). We know that the presence of moderate-to-severe PVL after TAVI is associated with increased mortality, but the significance of mild PVL after TAVI remains undefined [14]. The treatment of PVL depends on the severity and the consequences of PVL. In patients with significant PVL, balloon post-dilatation, valve-in-valve TAVI, percutaneous closure with a plug, surgical intervention or medical treatment should all be considered on a case-by-case basis.

### 4.2. New Pacemaker Implantation and New Left Bundle Branch Block (LBBB)

Even though, over the years, the incidence of new conduction abnormalities and pacemaker implantation has decreased, most studies still show a higher incidence of conduction abnormalities after TAVI as compared to SAVR, in particular for self-expanding valves, with a reported incidence of 17–25% for new pacemaker implantation in more recent trials [15,17] (Table 2). In the PARTNER 3 trial, there was no difference between the TAVI and SAVR groups regrading new pacemaker implantation, but the incidence of new left bundle branch block was higher in the balloon-expandable TAVI group as compared to the SAVR group (22% vs. 8%) [16]. The risk of conduction abnormalities and new pacemaker implantation is higher in the first 2 days after TAVI and is significantly increased in patients with baseline right bundle branch block, severe annular calcifications and a lower implant depth, whereas a higher deployment of the valve has been associated with a decreased risk of new conduction abnormalities after TAVI [18,19,20]. The data regarding the prognostic impact of new left-bundle branch block and pacemaker after TAVI are controversial. In the SURTAVI trial, survival at 1 year was not different in patients with a new pacemaker compared to the overall population, whereas in other studies, mortality was significantly increased in TAVI patients with a new pacemaker, in particular for pacemaker-dependent patients [15,21]. In a sub-analysis of the PARTNER 2 trial, new-onset LBBB was associated with significantly increased all-cause and cardiovascular mortality, hospitalization and pacemaker implantation [22]. These patients should be closely followed up and, in patients with a QRS duration of >150 ms and prolonged PR >240 msec, continuous ECG monitoring or electrophysiologic testing might be considered to guide the decision for pacemaker implantation [23].

### 4.3. Embolic Events 

Stroke is a feared and devastating complication, associated with increased mortality, cognitive impairment, important functional and social consequences, and high costs. Although the risk of most TAVI complications has decreased in the last 10 years, the risk of TAVI-related stroke has remained stable at an incidence of about 2%; however, this is slightly lower with the newer generation of valves, between 1.1–1.2% [16,17,24,25] (Table 2). Moreover, even in the absence of symptoms, most TAVI patients have defects identified on cerebral MRI that may be associated with the development of cognitive impairment [26]. TAVI-related stroke is mainly caused by the embolization of debris from the valve or the vasculature and is less often related to arrhythmia. The size of the debris is correlated to the size of the cerebral lesion. The risk of stroke is higher in women as compared to men; it is higher in the first days after TAVI, it is slightly lower in balloon-expandable than in self-expandable valves, and it is not related to the use of pre- or post-dilatation nor the anti-platelet or anticoagulant treatment used [16,17,24,25,26,27,28]. 

Cerebral embolic protection devices have been developed for capturing and removing embolic material during TAVI, with the hope of reducing periprocedural stroke. The most used device is the Sentinel cerebral embolic protection device (Boston Scientific). It consists of two filters within a single 6-French delivery catheter, which are placed percutaneously before TAVI, into the brachiocephalic artery (proximal filter) and the left common carotid artery (distal filter), using a right radial or brachial artery access. The use of Sentinel is safe, with a feasibility of >90% and a low rate of complications; however, although it has been shown to significantly reduce new ischemic brain lesions post-TAVI, there is no clear evidence proving a decrease in stroke incidence after TAVI [26,28]. The recent PROTECTED TAVR trial failed to show a significant difference in the incidence of stroke after TAVI in patients with and without the cerebral protection device, even if the incidence of disabling stroke was numerically lower. Whether the negative results of this trial are more related to the design of the trial than to the lack of effectiveness of the device is a matter of debate. The residual stroke risk may be related to smaller debris particles that may pass the filters, to an eventual malapposition of the filters or to embolization through the left vertebral artery, which is not covered [28]. Future studies, such as the BHF PROTECT-TAVI trial, will hopefully shed more light on the effectiveness of cerebral protection devices.

### 4.4. Vascular Complications

Access site-related vascular complications remain the most frequent complication after TAVI and are associated with worse short- and long-term outcomes. In the STS/ACC TVT registry, 9.6% of TAVI patients had a vascular complication, and 7.6% of patients had an access site bleeding event [29]. However, the incidence of access site-related complications has decreased over the years, owing to a decrease in the size of the sheaths and of the anti-thrombotic treatment used, the utilization of Doppler echocardiography for determining the best site for vascular puncture, and the use of percutaneous vascular closure devices. Prompt and efficient diagnoses and management are necessary for achieving bleeding control, which is usually carried out via crossover angiography from the contralateral femoral artery or, more recently, from the radial artery. Limited dissection or perforation can usually be managed with prolonged occlusive balloon inflation, whereas percutaneous deployment of a stent, thrombin injection or surgical repair can be used in cases with more extensive, flow-limiting dissection or bleeding, or in cases with hemodynamic instability or threatened limb circulation [30].

### 4.5. Valve Durability and Valve-in-Valve TAVR

Transcatheter valve durability remains one of the limiting aspects to the extension of TAVI in younger patients. Studies have shown that transcatheter valves, in particular supra-annular self-expandable valves, have lower gradients and higher effective orifice areas as compared to surgically implanted bioprostheses, values which are stable over 2 and up to 8 years of follow up, with low rates of structural valve deterioration or bioprosthetic valve failure, comparable to those seen in SAVR patients [14,15,16,17,31,32,33,34]. Although the data are encouraging, it should be pointed out that most of this evidence comes from older patients and cannot be extended to younger patients. In the current guidelines, the limiting age for considering TAVR is 75 years in the European guidelines, and 65 years of age in the American guidelines [1,2]. The patient’s comorbidities and the individual expected life expectancy as compared to the durability of the prosthesis should be taken into consideration in the decision making, but in the absence of evidence, SAVR remains the treatment of choice in young patients with severe AS and indication for intervention [1,2]. Although in our daily practice we see more and more and more young patients and patients at low surgical risk asking us about the possibility of performing TAVI, mainly related to the fear of the surgical intervention, TAVI should be strongly discouraged, and patients should be reassured and informed about the actual risks of the surgical intervention in their case. We should stress the higher risks of stroke, PVL and conduction disturbances related to TAVI as compared to SAVR, the absence or the limited data available in these groups of patients, and the risks related to a second intervention. Whenever a biological surgical or transcatheter valve is implanted in a younger patient, the risk of two or more interventions is high, and a careful life management plan should be considered [35]. Performing SAVR after TAVI is associated with a higher risk as compared to SAVR on a native valve; the resection of the prosthesis requires in most cases a more extensive surgery with associated root or ascending aorta replacement [35].

Valve-in-valve TAVI has emerged as an appealing, less invasive alternative to surgical reintervention in patients with bioprosthetic valve failure, being associated with significantly lower rates of 30-day morbidity and mortality, a lower risk of bleeding and a shorter hospitalization [35,36,37,38]. Valve-in-valve TAVI is, at the moment, the preferred treatment option in older or multiple-comorbidity patients with degenerated, surgically implanted or transcatheter bioprosthetic valves. However, valve-in-valve TAVI can be associated with higher gradients and higher rates of patient–prosthesis mismatch (in particular for small initial bioprostheses), as well as with a higher risk of acute coronary obstruction. The obstruction of a coronary artery is a feared complication of valve-in-valve TAVI that can occur in about 2 to 3% of patients [37]. Coronary artery obstruction can be caused by direct obstruction of the coronary ostia by the underling valve leaflets, pushed outward, or indirectly by sequestering the sinus of Valsalva at the sino-tubular junction. When a second prosthesis is implanted in a patient with a previous transcatheter valve, the leaflets of the first prosthesis are pushed open upwards, sealing the stent frame circumferentially up to the commissure level. If the commissure level of the first prosthesis is above the sino-tubular junction and its stent frame is in close proximity to the sino-tubular junction, the risk of coronary sinus sequestration with TAV-in-TAV is high [39]. Pre-procedural CCT plays an important role in evaluating of the risk of coronary obstruction before valve-in-valve TAVI and TAV-in-TAV. Coronary artery obstruction with valve-in-valve TAVI can have catastrophic implications and, whenever the risk of coronary artery obstruction with valve-in-valve TAVI estimated by the pre-procedural CT is high, surgery should be considered instead. Several reports have shown the feasibility of bioprosthesis leaflet laceration with an electrocautery wire (BASILICA) before valve-in-valve TAVI, in order to prevent acute coronary artery obstruction; however, the procedure is only limited to high specialized centers, and is not feasible in all cases [40].

### 4.6. Coronary Access after TAVI 

Many patients with AS have associated coronary artery disease, and about 10% of TAVI patients have an acute coronary syndrome in the first 2 years after TAVI, which is associated with a high mortality [41]. In general, the risk of difficult coronary artery access after TAVI is greater for supra-annular prostheses and with tall stent frames and small struts, but some studies have shown no significant differences between the type of prosthesis and the difficulty in obtaining coronary cannulation [42]. The incidence of unsuccessful coronary cannulation or unsuccessful PCI after TAVI varies between 3–7% in studies, to up to 35% of patients in real-world registry data, and the risk is higher for TAVI-in-TAVI procedures [42,43,44]. Maintaining good coronary access is particularly important for younger patients, and several strategies are available: implanting a valve with a sub-coronary frame position, obtaining commissural alignment for supra-annular valves and choosing prostheses with large open cells [35,45].

### 4.7. TAVI in Bicuspid Aortic Valve

Bicuspid aortic valves pose several challenges for TAVI, related to the often-asymmetrical aortic annulus, the presence of the raphe, which is often calcified, and the associated aortic root dilatation. Although studies with earlier prostheses have shown worse outcomes and a higher risk of PVL and aortic root injury, as compared to TAVI in tricuspid valves, more recent studies show no difference in the mortality and valve hemodynamics in TAVI in bicuspid vs. tricuspid aortic valves; however, the risk of significant PVL and stroke is higher [46,47]. There are little data about the anatomy of the bicuspid aortic valve that favors TAVI, the sizing of the valve and the best prosthesis for TAVI in bicuspid aortic valves. We need more data on the durability of TAVI in bicuspid aortic valves, on patient selection and on the sizing of the prosthesis. However, we know that TAVI in patients with severe and asymmetric leaflets and left ventricular outflow calcifications, with raphe calcifications, with a more elliptical aortic annulus or with a dilated ascending aorta >45 mm, can result in suboptimal prostheses expansions, and are associated with worse outcomes [47,48]. 

### 4.8. TAVI in Aortic Regurgitation

Aortic regurgitation also poses several challenges to the performance of TAVI, which are related to the larger annulus dimensions, the often-asymmetric annuli with a higher risk of PVL and the absence of valve calcifications, which are the landmark and the substrate for anchoring the prosthesis. Little evidence exists that shows good results in non-operable patients with pure aortic regurgitation and, according to the guidelines, TAVI may be considered in selected, non-operable patients with severe AR [1]. Newer valves have been developed specifically for patients with aortic regurgitation, such as the JenaValve (JenaValve Technology GmbH, Munich, Germany), which has a clip-based fixation over the native aortic leaflets. The ALIGN AR study is assessing the efficacy and the safety of the JenaValve system in patients with symptomatic severe aortic regurgitation who are at high surgical risk. 

## 5. Conclusions

TAVI has seen a remarkable evolution over the last 20 years, with an expansion of its use from non-operable to high- and intermediate-risk patients with severe AS. It has also become “less invasive” and in centers with expertise, a minimalist, “PCI-like” intervention can be performed in highly selected cases of transfemoral TAVI, with good results and a decrease in hospital stay and of costs related to the hospitalization. The rate of complications after TAVI has decreased overall, but the incidence of stroke, new pacemaker implantation and paravalvular leak remains higher compared to SAVR. We need more data on the long-term durability of transcatheter prosthesis and, at the current moment, we have little or no evidence for using TAVI in low-risk and young patients; SAVR remains indicated in these patients. Although the short- and mid-term hemodynamic results are good, with low rates of structural valve degeneration, the risks associated with a second intervention, in particular, a higher risk of patient–prosthesis mismatch, the higher risk of coronary obstruction and of difficult coronary access should be considered. Although TAVI is not the solution for all patients with severe AS, and it faces many challenges as well as many remaining open questions in the field of TAVI, it is without doubt that when looking at the past and the present, the future of TAVI remains bright.

## Figures and Tables

**Table 1 jcm-12-05645-t001:** Comparison between the pivotal studies on transcatheter heart valve intervention with regard to patients included, patient age, the type of transcatheter valve, follow-up duration and the primary outcome.

	High-Risk Patients		High-Intermediate-Risk Patients	Intermediate-Risk Patients		Low-Risk Patients	
	PARTNER 1A	US CoreValve High Risk	UK TAVI	PARTNER 2A	SURTAVI	PARTNER 3	EVOLUT
Number of patients	699	795	913	2032	1660	950	1468
Study population	symptomatic severe AS	severe AS with heart failure symptoms	symptomatic severe AS	symptomatic severe AS	symptomatic severe AS	severe AS with an indication for intervention	severe AS with an indication for intervention
Type of valve	Balloon-expandable	Self-expandable	balloon- expandable and self-expandable	Balloon-expandable	Self-expandable	balloon-expandable	Self-expandable
Patient median age, for the TAVI group (years)	83.6	83.1	81.1	81.5	79.9	73.3	74
Follow-up (years)	5	1	1	2	2	1	2
Primary endpoint	All-cause death	All-cause death	All-cause death	death from any cause or disabling stroke	death from any cause or disabling stroke	death, stroke, rehospitalization	death or disabling stroke
Result (with regard to the primary outcome)	TAVI non-inferior to SAVR	TAVI superior to SAVR	TAVI non-inferior to SAVR	TAVI non-inferior to SAVR	TAVI non-inferior to SAVR	TAVI superior to SAVR	TAVI non-inferior to SAVR

AS, aortic stenosis; TAVI, transcatheter aortic valve intervention; SAVR, surgical aortic valve replacement.

**Table 2 jcm-12-05645-t002:** Incidence of the main TAVI complications in studies with more recent transcatheter valves.

	High-Intermediate-Risk Patients		Intermediate-Risk Patients				Low Risk Patients			
	UK TAVI		PARTNER 2A		SURTAVI		PARTNER 3		EVOLUT	
	TAVI	SAVR	TAVI	SAVR	TAVI	SAVR	TAVI	SAVR	TAVI	SAVR
Stroke	2.4	2.3	5.5	6.1	3.4	5.6	0.6	2.4	3.4	3.4
PVL at least moderate	2.4	0.9	3.7	0.6	3.5	0.7	0.8	0	3.4	0.4
Mild PVL	43.7	12.3	22.5	2.8	28.3	NA	28.7	4.2	36	3
New pacemaker implantation	11	6.7	8.5	6.9	25.9	6.6	6.5	4	17.4	6.1
Major vascular complications	10.1	2.3	7.9	5	6	1.1	2.2	1.5	3.8	3.2
Aortic valve reintervention	2.2	1.1	1.4	0.6	2.8	0.7	0.6	0.5	0.7	0.6
Severe PPM	NA	NA	NA	NA	NA	NA	NA	NA	1.1	4.4
Coronary artery obstruction	NA	NA	0.4	0.6	0.2	0	0.2	0.7	0.9	0.4

Numbers represent % of patients. The incidence of stroke, at least moderate PVL, mild PVL, new pacemaker implantation, major vascular complications, severe PPM and coronary artery obstruction is reported at 30days, with the exception of the UK TAVI trial, when the incidence and the severity of PVL were reported at 6 weeks. The incidence of aortic valve reintervention is reported at the end of the study period. TAVI, transcatheter aortic valve intervention; SAVR, surgical aortic valve replacement; PVL, paravalvular leak; PPM, patient prosthesis mismatch.

## Data Availability

Not applicable.

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
