# Peer review of "TAVI after More Than 20 Years"

_jcm, 2023, doi:10.3390/jcm12175645_

Round 1

Reviewer 1 Report

Well written review considering all aspects of TAVI, giving an important update to the readers.

I think that the conclusions must be more adherent to what was reported in the text. Adfirming that the complications after TAVI must be analyzed deeper "before extending its use in younger patients", it very reductive.

Low surgical risk patients indipendently to age are better treated with surgery: low incidence of paravalvular leack, lower incident of new pacemaker, lower risk of subclinical embolism and stroke.

The results of Partner Trial are non corrisponding to real world results.

According with STS data base the incidence of stroke after SAVR is around 0.9%. In STS TAVAR database in spite of recently treated lower risk patients and using cerebral protection device, the incidence remains around 2%.

In the conclusion these aspects, in my understanding, have to be reppresented.

Author Response

Dear Reviewer,

I thank you very much for your comments and remarks, which I found very helpful and to the point.

-I think that the conclusions must be more adherent to what was reported in the text. Adfirming that the complications after TAVI must be analyzed deeper "before extending its use in younger patients", it very reductive.

                -I could not agree more. I must apologize, I had finished late, and I was quite tired.

I have changed the conclusions completely.

Low surgical risk patients indipendently to age are better treated with surgery: low incidence of paravalvular leack, lower incident of new pacemaker, lower risk of subclinical embolism and stroke.

The results of Partner Trial are non corrisponding to real world results.

According with STS data base the incidence of stroke after SAVR is around 0.9%. In STS TAVAR database in spite of recently treated lower risk patients and using cerebral protection device, the incidence remains around 2%.

  -Yes, I agree, and I don’t believe that just because of the one-year results of the PARTNER 3 trial, which also included hospitalization, TAVI should be used in low risk patients. I have stated this more clearly now in the text and in the conclusion. Reading it again, I agree that I didn’t write it toot clerae before.

The lower rate of certain complications seen in the PARTNER trial, as for example the very low stroke incidence, could be related among others to the performance of the procedure in centers with high expertise, a highly selected patient population, a more aggressive antithrombotic treatment and not per se related to TAVI. And yes, real-world registry data is different.

In the conclusion these aspects, in my understanding, have to be reppresented

- I have changed the conclusions completely.

I thank you once again for your time.

Kind regards,

Reviewer 2 Report

This review article covered almost significant issues in the present era of the TAVI procedure.

There were some points to be noted.

-This article included no table or figure at all.  A table showing the pivotal studies or a figure showing the type of TAVI valves is helpful to understand this article. 

-Sinus sequestration is an emerging term in the case of valve-in-valve TAVI, which explains the mechanism of coronary obstruction.  This word should be included in the text.

Some sentences are long to read, which makes readers challenging to understand. 

Author Response

Dear Reviewer,

I thank you very much for your comments and remarks, which I found very useful and instructive.

This review article covered almost significant issues in the present era of the TAVI procedure.

There were some points to be noted.

-This article included no table or figure at all.  A table showing the pivotal studies or a figure showing the type of TAVI valves is helpful to understand this article. 

                -Yes, I totally agree. It is always nice to show images of the devices. Unfortunately, I had tried on previous occasions to get the approval for using images on valves or other devices from the companies, but I didn’t receive any answer to my emails.

To your suggestion, I have done 2 tables, comparing some of the pivotal studies, the first one with regards to the patients included, the type of transcatheter valve and the primary outcome, the second one about the main complications that I talk about in the text. The resolution is not the best for the first one, but I believe that turning it should be better. I apologize I am not the best at formatting things, but I have it as an Excel file also.

-Sinus sequestration is an emerging term in the case of valve-in-valve TAVI, which explains the mechanism of coronary obstruction.  This word should be included in the text.

  -I thank you for your suggestion. I included the coronary sinus sequestration on the paragraph with coronary artery obstruction.

Comments on the Quality of English Language

Some sentences are long to read, which makes readers challenging to understand. 

  • Thank you, I will consider this also for the future and I have changed the long sentences.

I thank you once again for your time.

Kind regards,

Round 2

Reviewer 2 Report

This article was revised enough to be published.